# *MAVS* Genetic Variation Is Associated with Decreased HIV-1 Replication In Vitro and Reduced CD4^+^ T Cell Infection in HIV-1-Infected Individuals

**DOI:** 10.3390/v12070764

**Published:** 2020-07-16

**Authors:** Melissa Stunnenberg, Lisa van Pul, Joris K. Sprokholt, Karel A. van Dort, Sonja I. Gringhuis, Teunis B. H. Geijtenbeek, Neeltje A. Kootstra

**Affiliations:** Amsterdam Infection & Immunity Institute, Department of Experimental Immunology, University of Amsterdam, Amsterdam UMC, Meibergdreef 9, 1105 AZ Amsterdam, The Netherlands; m.stunnenberg@amsterdamumc.nl (M.S.); l.vanpul@amsterdamumc.nl (L.v.P.); joris.sprokholt@gmail.com (J.K.S.); k.a.vandort@amsterdamumc.nl (K.A.v.D.); s.i.gringhuis@amsterdamumc.nl (S.I.G.)

**Keywords:** human immunodeficiency virus 1, viral sensing, antiviral immunity, *MAVS* genetic variation, HIV-1 replication, viral load, immune activation, T cell-induced immunity

## Abstract

The mitochondrial antiviral protein MAVS is a key player in the induction of antiviral responses; however, human immunodeficiency virus 1 (HIV-1) is able to suppress these responses. Two linked single nucleotide polymorphisms (SNPs) in the *MAVS* gene render MAVS insensitive to HIV-1-dependent suppression, and have been shown to be associated with a lower viral load at set point and delayed increase of viral load during disease progression. Here, we studied the underlying mechanisms involved in the control of viral replication in individuals homozygous for this *MAVS* genotype. We observed that individuals with the *MAVS* minor genotype had more stable total CD4^+^ T cell counts during a 7-year follow up and had lower cell-associated proviral DNA loads. Genetic variation in *MAVS* did not affect immune activation levels; however, a significantly lower percentage of naïve CD4^+^ but not CD8^+^ T cells was observed in the *MAVS* minor genotype. In vitro HIV-1 infection of peripheral blood mononuclear cells (PBMCs) from healthy donors with the *MAVS* minor genotype resulted in decreased viral replication. Although the precise underlying mechanism remains unclear, our data suggest that the protective effect of the *MAVS* minor genotype may be exerted by the initiation of local innate responses affecting viral replication and CD4^+^ T cell susceptibility.

## 1. Introduction

Human immunodeficiency virus 1 (HIV-1) infection is characterized by a lack of protective immunity against the virus [1]. During HIV-1 infection, insufficient priming of naïve T cells occurs, which is partially explained by suboptimal functioning of dendritic cells (DCs) crucial in the induction of antiviral immunity [1,2,3,4,5,6]. DCs contain the ability to sense viral pathogen-associated molecular patterns (PAMPs) with pattern recognition receptors (PRRs) [7]. Various PRRs have the ability to recognize HIV-1-specific PAMPs such as carbohydrate structures (DC-SIGN), viral DNA (cGAS, IFI16) and viral RNA (RIG-I, DEAD-box helicase DDX3) [8,9,10,11,12,13,14,15]. PRR triggering induces innate antiviral responses, such as antiviral type I interferon (IFN) and cytokine responses, subsequently leading to induction of adaptive immunity via DC activation [16,17,18,19,20]. Viral RNA is sensed by sensors such as MDA5, RIG-I, and DDX3, of which the latter two play an important role in sensing of HIV-1 RNA [12,15,21,22]. RIG-I is responsible for sensing cytosolic genomic HIV-1 RNA, whereas DDX3 recognizes prematurely aborted HIV-1 RNA produced during transcription initiation of the provirus [12,15]. The mitochondrial antiviral protein MAVS signals downstream of RIG-I and DDX3 and serves as a platform for TBK1/IKKɛ activation, thus containing the potency to elicit antiviral type I IFN and cytokine responses needed to combat HIV-1 infection [23,24].

For MAVS-dependent activation of IRF3 and NF-κB, the binding of TRAF3 to MAVS is crucial. However, HIV-1 is able to block MAVS-dependent signaling via polo-like kinase 1 (PLK1) that is able to anchor to MAVS. The MAVS-PLK1 interaction leads to ultimate impediment of the recruitment of TRAF3 to MAVS and thus MAVS-induced type I IFN and cytokine responses [15,25,26]. We have previously identified two linked single nucleotide polymorphisms (SNPs) in the *MAVS* gene (rs7262903 and rs7269320) which result in two amino acid substitutions Gln198Lys (Q198K) and Ser409Phe (S409F) that render the protein insensitive to the PLK1-dependent suppression by HIV-1, and result in robust antiviral type I IFN responses and a decrease of viral infection in DCs in vitro [15,27]. Individuals homozygous for the minor alleles rs7262903 and rs7269320 (minor genotype) are observed at a frequency of 2% in the population [15]. Interestingly, genome-wide association (GWA) data from the Amsterdam Cohort Studies strongly suggest that in untreated HIV-1-infected men who have sex with men (MSM), this genotype is associated with lower viral load in plasma at set point. Moreover, the minor genotype shows a delayed increase of viral load over the course of infection compared to the major genotype [15]. These data indicate that the MAVS pathway is important in controlling HIV-1 infection.

HIV-1 infection is characterized by continuous high levels of immune activation indicative of tissue damage and cell death due to continuous HIV-1 replication, co-infections with other pathogens, bacterial translocation or immune dysregulation [28,29,30,31,32]. HIV-1-specific cytotoxic T cell (CTL) responses are a strong correlate of viral control during the asymptomatic period of HIV-1 infection [33,34,35,36,37]. Although the breadth and magnitude of these responses are limited, the antiviral activity of these responses is associated with initial viral control and rapid selection of escape variants [38,39,40]. During the asymptomatic phase of infection, new T cell responses that target HIV escape variants increase in breadth, but eventually, the control of viremia is lost due to T cell dysfunction and viral escape [33,41,42,43].

Here, we investigated the underlying mechanism responsible for the effect of this *MAVS* genetic variation on the control of HIV-1 infection. We determined whether immune activation and CTL activity are associated with the protective effect of the *MAVS* minor genotype during the asymptomatic phase of infection. In addition, we analyzed the effect of *MAVS* genetic variation during viral replication in vitro.

Our data demonstrated that untreated HIV-1-infected individuals carrying the *MAVS* minor genotype had more stable CD4^+^ T cell counts during a 7-year follow up and a lower cell-associated proviral DNA load, as compared to individuals with the *MAVS* major genotype. Although the *MAVS* minor genotype was not associated with changes in immune activation levels, T cell exhaustion, activation, senescence, or HIV-1-specific cytokine production, we found a decreased percentage of naïve CD4^+^ T cells in individuals with the *MAVS* minor genotype. Furthermore, we observed that in vitro infection of peripheral blood mononuclear cells (PBMCs) from healthy donors with the *MAVS* minor genotype resulted in lower levels of HIV-1 replication. Although the underlying mechanism remains unclear, our data suggest that the protective effect of the *MAVS* minor genotype is not due to an intrinsic factor in CD4^+^ T cells that decreases HIV-1 replication, but might be due to factors released in the PBMC culture that inhibit HIV-1 replication in CD4^+^ T cells.

## 2. Materials and Methods

### 2.1. Study Participants

Study subjects participated in the Amsterdam Cohort Studies (ACS) on HIV-1 infection and AIDS among homosexual men and (injecting) drug users [44]. Untreated HIV-1 positive participants with known date of HIV-1 seroconversion, genotyped for the rs7262903 and rs7269320 SNPs and with sample availability were selected for analysis. Study participants with a *MAVS* major genotype were “typical” progressors and had a CD4^+^ T cell count <200/µL blood around 5–6 years post seroconversion. Selected study participants were negative for HLA*B27, HLA*B57, and the 32-base-pair deletion in CCR5. In this study, eight HIV-1-infected participants homozygous for the minor alleles rs7262903 and rs7269320 (*MAVS* minor genotype) and 37 HIV-1-infected individuals homozygous for the major alleles (*MAVS* major genotype) participated.

### 2.2. Ethics Statement

Healthy controls were obtained from the Dutch national blood bank (Sanquin, Amsterdam, The Netherlands). This study has been conducted in accordance with the ethical principles set out in the declaration of Helsinki and was approved by the institutional review board of the Academic Medical Center (AMC, Amsterdam, Netherlands) and the Ethics Advisory Body of the Sanquin Blood Supply Foundation (Amsterdam, Netherlands, MEC 07/182, approved on 20 August 2007). Written informed consent was obtained from all participants.

### 2.3. Quantification of Proviral DNA

CD4^+^ T cells of study participants were isolated using CD4 MACS Microbeads (Miltenyi Biotec, Bergisch Gladbach, Germany). DNA was then isolated using the AllPrep DNA/RNA kit (Qiagen, Hilden, Germany) and DNA concentrations were determined by nano-drop. The cell-associated proviral DNA load in the CD4^+^ T cells was determined by qPCR using the following primer set: HIV-pol-B (Fw) 5′-TAACCTGCCACCTGTAGTAGCAAAAGAAAT-3′ and Pol-E (Rev) 5′-ATGTGTACAATCTAGTTGCCA-3′. Reactions were carried out in 10 µL total volume containing 5 µL GoTaq qPCR Master Mix 2× (Promega, Madison, WI, USA); 2.5 µL DNA (10 ng/µL); 0.2 µL each of (20 µM) Forward and Reverse primer and 2.1 µL water. Amplification conditions consisted of a pre-incubation stage at 95 °C for 3 min, a pre-amplification stage consisting of two steps, 15 s at 95 °C and 15 s at 49 °C (2 cycles), amplification stage consisting of 3 steps 10 s at 95 °C, 20 s at 58 °C, 30 s at 72 °C (40 cycles), a melting curve stage consisting of 5 s at 95 °C and 1 min at 55 °C, and finally a cooling stage of 10 s at 40 °C. Reactions were performed using a LightCycler 480 (Roche, Basel, Switzerland). To quantify the HIV DNA copy number, the 8E5/LAV cell line was used to generate a standard curve [45]. When the proviral load was below the detection limit of the qPCR, a single genome amplification (SGA) detecting HIV-pol was performed using the following primer sets: primary PCR HIV Pol-F (Fw) 5′- TTAGTCAGTGCTGGAATCAGG-3′ and Pol-D (Rev) 5′-GCTACATGAACTGCTACCAGG-3′, and nested PCR; Pol-B and Pol-E. At least 10 PCR reactions using GoTaq polymerase (Promega, Madison, WI, USA) were performed per sample using an input of 2.5–250 ng DNA. The PCR cycles for both primary and nested PCR reactions were as follows: denaturation 5 min at 94 °C followed by 35 cycles of 15 s at 94 °C, 30 s at 50 °C, 45 s at 72 °C and finally followed by 1 cycle of 5 min at 72 °C. Positive PCR reactions were visualized using a 1% agarose gel. When at least one third of the PCR reactions was negative, it was assumed that each PCR reaction contained no more than 1 copy of proviral DNA.

### 2.4. Multiplex Assay

Cytokine and chemokine levels in serum were analyzed for expression levels of TNF-α, IL-1β, IL-2, IL-4, IL-6, IL-10, IP-10, MCP-1, Mip1α, and Mip1β using a Bio-Plex Pro Human Cytokine 10-plex (Bio-rad, Hercules, CA, USA) according to manufacturer’s instructions. IFN-α, IFN-β, IFN-γ, IL-27p28, and IL-12p70 levels were assessed using Bio-Plex Pro Human Inflammatory 5-plex (Bio-rad, Hercules, CA, USA). Measurements were performed in duplicates and read using the Bio-Plex 200 system (Bio-rad, Hercules, CA, USA).

### 2.5. T Cell Phenotyping

Cryopreserved peripheral blood mononuclear cells (PBMCs) were stained with monoclonal antibodies for 30 min at 4 °C to perform T cell phenotyping using flow cytometry. For this study monoclonal and directly conjugated antibodies were used: CD3 (V500, 561416, BD Biosciences, San Jose, CA, USA), CD4 (BV650, 563875, BD Biosciences, San Jose, CA, USA), CD8 (BV785, 301046, Biolegend, San Diego, CA, USA), CD45 (APC, 304150, Biolegend, San Diego, CA, USA), PD1 (BB515, 564494, BD Biosciences, San Jose, CA, USA), LAG-3 (PE, 12-2239-42, Thermo Fisher, Waltham, MA, USA), CD27 (APC-FIRE 750, 302846, Biolegend, San Diego, CA, USA), CD28 (APC-R700, 565181, BD Biosciences, San Jose, CA, USA), CD38 (BV421, 303526, Biolegend, San Diego, CA, USA), HLA-DR (BB700, 745782, BD Biosciences, San Jose, CA, USA), CD134 (PE-Cy7, 563663, BD Biosciences, San Jose, CA, USA), and CCR7 (BUV395, 563977, BD Biosciences, San Jose, CA, USA), and cell viability was assessed (LIVE/DEAD Fixable Red Dead Cell Stain Kit, L34972, Thermo Fisher, Waltham, MA, USA). Cells were fixed using Cell fix (BD Biosciences, San Jose, CA, USA) and flow cytometric analysis was performed on a BD LSR II/Fortessa flow cytometer (BD Biosciences, San Jose, CA, USA) and analyzed using FlowJo software v10 (TreeStar, Ashland, OR, USA). The percentage of PD1^+^LAG-3^+^ cells, CD27^−^CD28^−^ cells and HLA-DR^+^CD38^+^ and/or CD134^+^ cells within CD4^+^ and CD8^+^ T cell populations was assessed and indicative of T cell exhaustion, senescence and activation, respectively. T cell differentiation was characterized within live CD4^+^ and CD8^+^ T cells and comprised naïve (T_n_; CD45RA^+^CD27^+^CCR7^+^), effector memory (EM; CD45RA^−^CCR7^−^CD27^−^), transitional memory (TM; CD45RA^−^CCR7^−^CD27^+^), central memory (CM; CD45RA^−^CCR7^+^CD27^+^), and terminally differentiated effector memory (TEMRA; CD45RA^+^CCR7^−^CD27^−^).

### 2.6. Intracellular Cytokine Production of HIV-1-Specific T Cells

Cryopreserved PBMCs were thawed, treated with 20 U RQ1 RNAse-free DNAse (Promega, Madison, WI, USA) and kept at 37 °C 10% CO_2_. After O/N resting, PBMCs were washed, seeded in RPMI 1640 (Thermo Fisher, Waltham, MA, USA) supplemented with 10% FCS (Sigma Aldrich, St. Louis, MO, USA), 10 U/mL penicillin (Thermo Fisher, Waltham, MA, USA), 10 mg/mL streptomycin (Thermo Fisher, Waltham, MA, USA) and 2 mM l-glutamine (Lonza, Basel, Switzerland) and stimulated with 2 µg/mL anti-CD28 and 1 µg/mL anti-CD29 (both obtained from Sanquin, Amsterdam, The Netherlands) for co-stimulation. Stimulations occurred in the presence or absence of HIV-1 consensus B Gag peptide pool (2 µg/mL, 12425, NIH AIDS reagents, Germantown, MD, USA) to activate HIV-1 specific T cells or in the presence of Staphylococcus aureus enterotoxin B (SEB, 10 pg/mL, Sigma Aldrich, St. Louis, MO, USA), as a positive control. Furthermore, cells were treated with Brefeldin A and GolgiStop (BD Biosciences, San Jose, CA, USA), according to manufacturer’s instructions and incubated in the presence of an antibody directed against CD107a to detect degranulation (FITC, 11-1079-42, Thermo Fisher, Waltham, MA, USA), for 6 h. After 6 h, cells were washed and stained with monoclonal antibodies for 30 min at 4 °C to assess expression levels of surface markers CD3 (V500, 561416, BD Biosciences, San Jose, CA, USA), CD4 (BUV395, 563550, BD Biosciences, San Jose, CA, USA), CD8 (Pacific Blue, 301023, Biolegend, San Diego, CA, USA) and CD137 (BV650, 564092, BD Biosciences, San Jose, CA, USA), and viability was assessed (LIVE/DEAD Fixable Red Dead Cell Stain Kit, L34972, Thermo Fisher, Waltham, MA, USA). Then, cells were washed, permeabilized using a Cytofix/Cytoperm kit (BD Biosciences, San Jose, CA, USA) and stained for 30 min at 4 °C to determine the expression levels of intracellular cytokines IL-2 (PE, 340450, BD Biosciences, San Jose, CA, USA), TNF-α (Alexa Fluor 700, 557996, BD Biosciences, San Jose, CA, USA), Mip-1β (PE-Cy7, 560687, BD Biosciences, San Jose, CA, USA), and IFN-γ (APC-Alexa Fluor 750, MHCIFG27, Thermo Fisher, Waltham, MA, USA). Cells were fixed using Cell fix (BD Biosciences, San Jose, CA, USA) and intracellular cytokine levels were assessed in viable CD3^+^, CD4^+^, and CD8^+^ T cells using flow cytometric analysis on a BD LSR II/Fortessa flow cytometer (BD Biosciences, San Jose, CA, USA) and analyzed using FlowJo software v10 (TreeStar, Ashland, OR, USA).

### 2.7. Cell Isolation and Infection

PBMCs were obtained from buffy coats of healthy blood donors (Sanquin, Amsterdam, The Netherlands) by Ficoll-Isopaque density gradient centrifugation. *MAVS* genetic variation was assessed by screening for SNPs rs7262903 and rs7269320 using TaqMan genotyping Assays (ID C_25623847 and C_25623845, Thermo Fisher, Waltham, MA, USA), according to the manufacturer’s instructions. Monocytes were isolated from PBMCs using an additional percoll density gradient. Monocyte-derived DCs (mDCs) were generated as previously described [46]. On day 6, mDCs were treated with 90% methanol and stained using an anti-MAVS antibody (3993S, Cell Signaling, Danvers, MA, USA) followed by PE-conjugated donkey anti-rabbit (Jackson Immuno Research, West Grove, PA, USA). MAVS expression was measured with flow cytometry using the FACS Canto II (BD Biosciences, San Jose, CA, USA) and analyzed with FlowJo software v10 (TreeStar, Ashland, OR, USA). CD4^+^ T cells were isolated from PBMCs using CD4 MACS Microbeads (Miltenyi Biotec, Bergisch Gladbach, Germany), according to the manufacturer’s instructions. PBMCs or CD4^+^ T cells were stimulated for 3 days in Iscove’s modified Dulbecco’s medium (IMDM, Thermo Fisher, Waltham, MA, USA) supplemented with 10% fetal bovine serum, penicillin (100 U/mL, Thermo Fisher, Waltham, MA, USA), streptomycin (100 U/mL, Thermo Fisher, Waltham, MA, USA), and recombinant IL-2 (20 U/mL; Novartis, Basel, Switzerland) in the presence or absence of phorbol myristate acetate (PMA, 10 ng/mL, Sigma Aldrich, St. Louis, MO, USA) and ionomycin (1 mg/mL, Sigma Aldrich, St. Louis, MO, USA) at a cell density of 5 × 10^6^/mL (PBMCs) and 1 × 10^6^/mL (CD4^+^ T cells). After 3 days, PBMCs and CD4^+^ T cells were infected with a single round VSV-G-pseudotyped HIV-1 containing a fire-fly luciferase gene [47] at a multiplicity of infection (MOI) of 0.5 or with R5-tropic strain NL4.3 BaL (MOI 0.1) or X4-tropic strain NL4.3 (MOI 0.1), and cultured for 3 and 8 days at a cell density of 1 × 10^6^/mL. Viral replication was determined using different assays: Infection by single round VSV-G-pseudotyped HIV-1 containing a fire-fly luciferase gene was determined after 72 h, by measuring luciferase activity (luminescence in relative light units (RLUs)) using a luciferase substrate (0.83 mM ATP, 0.83 mM d-luciferin (Duchefa, Haarlem, The Netherlands), 18.7 mM MgCl_2_, 0.78 µM Na_2_H_2_P_2_O_7_, 38.9 mM Tris pH 7.8, 0.39% (*v*/*v*) glycerol, 0.03% (*v*/*v*) Triton X-100, and 2.6 µM dithiothreitol). Viral replication of NL4.3 BaL and NL4.3 was determined after 8 days by Gag p24 production in the PBMC culture supernatant using a quantitative in-house p24 ELISA or by intracellular staining. In brief, cells were fixed with 4% paraformaldehyde (PFA) and stained intracellularly using a directly conjugated anti-p24 (PE, KC57 RD-1, Beckman Coulter, Brea, CA, USA). Intracellular Gag p24 levels were assessed using flow cytometry and analyzed using the FACS Canto II (BD Biosciences, San Jose, CA, USA) and FlowJo software v10 (TreeStar, Ashland, OR, USA).

### 2.8. Statistical Analysis

Statistics were performed using a Mann–Whitney test for unpaired observations and a Wilcoxon test for paired observations, as indicated, using GraphPad version 8 (San Diego, CA, USA). Statistical significance was set at *p* < 0.05.

## 3. Results

### 3.1. MAVS Minor Genotype Is Associated with Sustained CD4^+^ T Cell Counts and a Reduced Proviral DNA Load during HIV-1 Infection

Individuals carrying a *MAVS* minor genotype have lower viral loads in plasma at set point and show a delayed increase of viral load as compared with individuals with the *MAVS* major genotype [15]. Here, we examined the levels of CD4^+^ and CD8^+^ T cells in individuals with a *MAVS* minor (*n* = 6) and *MAVS* major (*n* = 7) genotype, at 3, 5, and 7 years post seroconversion (p. SC). Individuals with the *MAVS* major genotype showed significantly lower CD4^+^ T cell counts 7 years p.SC compared with 3 years p.SC, whereas in individuals with the *MAVS* minor genotype no significant change in CD4^+^ T cell counts was observed over time (Figure 1a). CD8^+^ T cell counts remained similar over time in both *MAVS* genotype groups (Figure 1b). We next determined the cell-associated viral DNA levels in CD4^+^ T cells in individuals carrying a *MAVS* minor or *MAVS* major genotype at a time point with similar CD4^+^ T cell counts (3.5–8.5 years p.SC; *n* = 7) (Figure 1c). We observed that CD4^+^ T cells obtained from individuals carrying the *MAVS* minor genotype contained significantly lower levels of cell-associated proviral DNA compared with those with a *MAVS* major genotype (Figure 1d). These data strongly suggest that the protective effect of the *MAVS* minor genotype is associated with more stable CD4^+^ T cell counts and lower levels of HIV-1-infected CD4^+^ T cells during the course of disease.

### 3.2. MAVS Genetic Variation Does Not Affect Immune Activation Levels

Subsequently, we studied the level of immune activation in serum of individuals carrying a *MAVS* minor or *MAVS* major genotype, using a multiplex assay. Immune activation levels were measured at two different time points during infection: early p.SC (2.5–3.5 years) and late p.SC (4.5–5.5 years). The levels of IFN and interferon-stimulated genes (ISGs) measured by protein levels of IFN-α, IFN-β, IFN-γ and IL-27p28 did not significantly change over time in the minor (*n =* 8) as well as the major genotype (*n =* 8) and no differences were observed between *MAVS* genotypes (Figure 2a). The IFN-γ-inducible protein IP-10 (CXCL-10) was significantly increased in *MAVS* minor individuals (*n =* 8) over time, and a similar trend was observed in *MAVS* major individuals (*n =* 8 *MAVS* minors; *n =* 37 *MAVS* majors). Furthermore, the levels of several pro-inflammatory cytokines (TNF-α, IL-1β, IL-12p70, IL-6), regulatory cytokines (IL-2, IL-4), anti-inflammatory cytokines (IL-10) and chemokines (MCP-1, Mip-1α and Mip-1β, or known as CCL2, CCL3 and CCL4, respectively) did not differ between the different *MAVS* genotypes early p.SC (*n =* 8) and late p.SC (*n* = 8 *MAVS* minors; *n* = 37 *MAVS* majors for all proteins except IL-12p70) (Figure 2b,c). These data indicate that *MAVS* genetic variation does not affect immune activation levels in serum.

### 3.3. MAVS Minor Genotype Contains a Decreased Percentage of Naïve CD4^+^ T Cells

Next, we analyzed the percentage of CD4^+^ and CD8^+^ T cells positive for the activation and exhaustion markers PD-1, LAG-3, CD38, HLA-DR and CD134 (OX40) of individuals carrying the *MAVS* minor and *MAVS* major genotypes (2.5–3.5 years p.SC; *n =* 4), using flow cytometry. No differences were observed between the different cell populations between *MAVS* genotypes (Figure 3a). Then, we investigated immune senescence by measuring percentages of CD27^−^CD28^−^ CD4^+^ T cells and observed that no difference was detected in the percentage of senescent CD4^+^ T cells between the different *MAVS* genotypes (Figure 3a). In the CD8^+^ T cell population, the percentage of PD-1^+^, LAG-3^+^, PD-1^+^LAG3^+^, CD38^+^, HLA-DR^+^, CD38^+^HLA-DR^+^, CD134^+^, or senescence (CD27^−^CD28^−^) did not differ between *MAVS* genotypes (Figure 3b). Next, we examined whether *MAVS* genetic variation affects T cell differentiation as reflected by the percentages of different CD4^+^ T cell populations comprised of naïve T cells (T_n_; CD45RA^+^CD27^+^CCR7^+^) and differentiated phenotypes defined as terminally differentiated effector memory cells (TEMRA; CD45RA^+^CCR7^−^CD27^−^), central memory (CM; CD45RA^−^CCR7^+^CD27^+^), transitional memory (TM; CD45RA^−^CCR7^−^CD27^+^) and effector memory (EM; CD45RA^−^CCR7^−^CD27^−^) T cells (*n =* 4). We observed that individuals carrying the *MAVS* minor genotype contained a significantly lower percentage of naïve CD4^+^ T cells. Although the *MAVS* minor genotype showed an increased percentage of effector/memory CD4^+^ T cells, the individual percentages of CD4^+^ TEMRA, CM, TM and EM populations did not differ between *MAVS* genotypes (Figure 4a). In contrast, the percentage of CD8^+^ naïve, TEMRA, CM, TM and EM did not differ between *MAVS* genotypes (Figure 4a). In addition, absolute CD4^+^ T cell counts showed a trend towards decreased naïve CD4^+^ T cells (*p* = 0.057), which was not observed in naïve CD8^+^ T cells (Figure 4b). The individual CD4^+^ T cell subsets did not significantly differ in number between the different *MAVS* genotypes (Figure 4b). Furthermore, the percentages of exhausted and activated T cells within the different T cell subpopulations did not differ between *MAVS* genotypes (Figure 4c,d). Although *MAVS* genetic variation did not affect CD4^+^ and CD8^+^ T cell exhaustion, activation and senescence, our data suggest that individuals carrying the *MAVS* minor genotype contain lower percentages of naïve CD4^+^ T cells.

### 3.4. MAVS Genetic Variation Does Not Affect Intracellular Cytokine Responses of HIV-1-Specific T Cells

To determine the effect of *MAVS* genetic variation on HIV-1-specific T cell cytokine responses, we assessed the intracellular expression levels of pro-inflammatory and regulatory cytokines in CD4^+^ and CD8^+^ T cells of individuals carrying a *MAVS* minor or major genotype (*n =* 4 and *n =* 3, respectively). Peripheral blood mononuclear cells (PBMCs) obtained 2.5–3.5 years p.SC were stimulated with an HIV-1 consensus B Gag peptide pool to achieve activation of HIV-1-specific T cells. Staphylococcal enterotoxin B (SEB) was used as a positive control. The percentages of CD4^+^ and CD8^+^ T cells positive for TNF receptor family member CD137, CD107a, and intracellular cytokines were analyzed using flow cytometry. CD137 functions as a costimulatory molecule important for activation of virus-specific memory T cell responses and CD107a is indicative of T cell degranulation [48,49]. Here, the percentages of CD137 and CD107a on CD4^+^ T cells were not affected by Gag stimulation, whereas SEB-treatment resulted in enhanced CD137 expression and degranulation (Figure 5a). Furthermore, *MAVS* genetic variation did not affect the percentage of CD137^+^ and CD107a^+^ CD4^+^ T cells. Next, we examined percentages of CD4^+^ T cells positive for intracellular cytokines IFN-γ, IL-2, Mip-1β and TNF-α. We observed that the percentages of IFN-γ^+^ and TNF-α^+^ CD4^+^ T cells were not enhanced upon Gag stimulation, irrespective of *MAVS* genetic variation (Figure 5a), whereas, as expected, SEB-treated CD4^+^ T cells showed enhanced cytokine responses. Upon Gag stimulation, the percentage of Mip-1β^+^ CD4^+^ T cells increased, albeit to a similar extent in both *MAVS* genotypes. The intracellular cytokine levels did not differ between the different *MAVS* genotypes (Figure 5a). Similar flow cytometric analyses were performed in CD8^+^ T cells (Figure 5b). The percentage of CD137^+^ and CD107a^+^ CD8^+^ T cells and intracellular cytokines was similar between different *MAVS* genotypes. Our results imply that *MAVS* genetic variation does not affect intracellular cytokine production of CD4^+^ and CD8^+^ HIV-1-specific T cells upon in vitro stimulation.

### 3.5. MAVS Minor Genotype Affects HIV-1 Replication In Vitro

Next, we investigated whether *MAVS* genetic variation affects HIV-1 replication in vitro. First, we assessed the effect of the SNP in *MAVS* on MAVS protein expression levels. Intracellular MAVS protein expression levels were analyzed in monocyte-derived DCs from healthy donors using flow cytometry and did not differ between the genotypes (Figure 6a). Next, PBMCs from healthy blood donors with different *MAVS* genotypes were infected with single round VSV-G-pseudotyped HIV-1 containing a fire-fly luciferase gene to quantify viral protein production. Even though virus replication in PBMCs with a *MAVS* major and heterozygous genotype was similar, viral replication levels were decreased in PBMCs of individuals with the *MAVS* minor genotype (Figure 6b). Furthermore, PBMCs were infected with R5-tropic strain NL4.3 BaL. Notably, viral replication levels in PBMCs from individuals with the *MAVS* minor genotype were significantly lower than observed in PBMCs from the *MAVS* major and *MAVS* heterozygous genotype (Figure 6c). To examine whether the difference in HIV-1 replication levels in PBMCs is due to an intrinsic factor in CD4^+^ T cells with the *MAVS* minor genotype, CD4^+^ T cells from healthy donors with the different *MAVS* genotypes were infected with R5-tropic strain NL4.3 BaL or X4-tropic strain NL4.3. Although infection levels were low, no significant difference in intracellular p24 levels between the genotypes was observed (Figure 6d). Moreover, no differences in viral replication levels between the genotypes were observed when phorbol myristate acetate (PMA)-ionomycin-activated CD4^+^ T cells were infected with NL4.3 (Figure 6e). In addition, CD4^+^ T cells were infected with VSV-G-pseudotyped HIV-1 and viral replication as determined by luciferase activity was assessed and did not differ between the different *MAVS* genotypes (Figure 6f). These data indicate that *MAVS* genetic variation affects HIV-1 replication in vitro in the PBMC culture, while no intrinsic restriction to infection was observed in CD4^+^ T cells.

## 4. Discussion

HIV-1 has developed various ways to evade immune surveillance, one of which is illustrated by PLK1-dependent evasion of MAVS-mediated induction of antiviral type I IFN and cytokine responses [15]. We previously identified two linked SNPs in the *MAVS* gene that render the MAVS protein insensitive to the PLK1-dependent HIV-1 evasion mechanism and it was shown that untreated HIV-1-infected individuals homozygous for the minor alleles (*MAVS* minor genotype) contained a decreased viral load at set point and prolonged control of viral replication during the course of HIV-1 infection [15]. Here, we investigated whether innate and adaptive immune responses contribute to the protective effect of the *MAVS* minor genotype by comparing the immune activation levels and T cell phenotype and function with responses induced in individuals carrying the *MAVS* major genotype. We observed that individuals with the *MAVS* minor genotype showed no significant decrease of CD4^+^ T cells counts during a 7-year follow up. Moreover, individuals homozygous for the *MAVS* minor genotype had lower levels of CD4^+^ T cell-associated proviral DNA, as compared with individuals with the *MAVS* major genotype with comparable CD4^+^ T cell counts. The levels of immune activation, T cell exhaustion, activation, senescence, and intracellular cytokine responses of HIV-1-specific T cells did not differ between individuals with the different *MAVS* genotypes. However, we did observe a significantly lower percentage of naïve CD4^+^ T cells in HIV-1-infected individuals with the *MAVS* minor genotype. Furthermore, in vitro infection of PBMCs but not CD4^+^ T cells of healthy donors carrying the *MAVS* minor genotype resulted in decreased levels of viral replication. Our data suggest that the protective effect of the *MAVS* minor genotype is associated with control of HIV-1 infection in CD4^+^ T cells in the PBMC culture, rather than by an intrinsic restriction in CD4^+^ T cells.

Immune activation levels are a strong correlate of HIV-1 disease progression [30,31,32]. Here, we analyzed whether the protective effect of the *MAVS* minor genotype could attribute to changes in the level of immune activation analyzed by cytokine/chemokine levels and T cell activation. Cytokine and chemokine levels in serum as well as the expression of CD38 and HLA-DR on CD4^+^ and CD8^+^ T cells did not significantly differ between the different *MAVS* genotypes early and late after seroconversion. Moreover, no differences in T cell exhaustion and senescence were observed between the genotypes. It should be noted that our study population consisted of typical progressors, in which HIV-1-infected individuals with rapid progression and CD4^+^ T cell counts below 200 cell/µL before 5 years after seroconversion were not analyzed. Strikingly, we observed that the IFN-γ-induced chemokine IP-10, which functions as a chemoattractant for NK cells, monocytes and T cells [50], was slightly increased in serum of *MAVS* minor individuals both early and late after seroconversion. IP-10 is considered one of the most important biomarkers for disease progression in (untreated) HIV-1 infection and macrophages in the gut are considered as the main source of circulating IP-10 [51,52,53]. However, IP-10 can also be produced by other cell types including T cells, monocytes, and monocyte-derived dendritic cells (mDCs) in the circulation and lymph nodes [54,55]. Here, the observed slight elevation of IP-10 in *MAVS* minor individuals who control HIV-1 infection, may be a reflection of the induction of an antiviral type I IFN and proper innate immune response.

To determine whether the protective effect of *MAVS* genetic variation could be explained by divergent T cell differentiation levels, we analyzed the percentages of naïve, TEMRA, CM, TM and EM CD4^+^, and CD8^+^ T cells. Interestingly, we found that individuals with the *MAVS* minor genotype contained lower percentages of naïve CD4^+^ T cells, and thus an increased total memory/effector CD4^+^ T cell population. The CD4^+^ memory T cells are the main population infected by HIV-1 [56,57], and depletion of this population during HIV-1 infection is instigated either directly via productive infection leading to caspase-3-dependent apoptosis or indirectly, via non-productive infection of bystander CD4^+^ T cells, resulting in caspase-1-dependent pyroptosis [58,59]. The observation that CD4^+^ T cells of individuals with the *MAVS* minor genotype harbored low levels of cell-associated proviral DNA may be indicative of lower virus-induced cell death and preservation of the memory CD4^+^ T cell population in the individuals with the *MAVS* minor genotype. Loss of the naïve CD4^+^ T cells in individuals during progressive HIV infection is caused by increased immune activation, high T cell turnover rates, and emergence of CXCR4-using HIV-1 variants [30,56]. In this study, individuals with the *MAVS* minor genotype showed no increase in immune activation, as compared with individuals with the *MAVS* major genotype, moreover no CXCR4-using HIV-1 variants have been detected in this group. It is therefore more likely that the observed decrease of the naïve CD4^+^ T cell population is the result of the preservation of the total memory/effector CD4^+^ T cell population.

HIV-1 specific CTL responses are a strong correlate of viral control during HIV-1 infection [34,60]. Studies have convincingly described the role of CTLs in the control of viremia in SIV infections of rhesus macaques, as well as in humans [60,61,62,63]. Although HIV-1 is able to escape immune pressure due to the occurrence of escape mutations in CTL epitopes [40,64,65], several studies imply that some escape mutations contain the ability to elicit continuous immune pressure on the virus, which can lead to virus attenuation [66,67,68]. Furthermore, disease progression is associated with HLA types and it has been demonstrated that HLA types that have been shown to correlate to low viremia favored responses targeting the conserved Gag protein [69,70,71,72,73,74]. Here, we examined functionality of HIV-1-specific CD4^+^ and CD8^+^ T cells, by using a Gag peptide pool to stimulate PBMCs. We did not observe an effect of *MAVS* genetic variation on the production of antigen-specific CD4^+^ and CD8^+^-induced cytokine and chemokine production. These data suggest that the protective effect of *MAVS* is not elicited by antigen-specific T cell functionality.

Interestingly, while MAVS protein levels were unaffected by *MAVS* genetic variation, in vitro infection of PBMCs from healthy individuals with a *MAVS* minor genotype resulted in decreased viral replication levels, indicating that these cells are less susceptible to HIV-1 infection and replication. Of note, *MAVS* genetic variation did not affect viral replication levels in purified CD4^+^ T cells. Considering the different ways of viral entry exploited by VSV-G-pseudotyped HIV-1 and R5-tropic virus NL4.3 BaL, viral replication in PBMCs with the *MAVS* minor genotype is restricted at a post-entry level. Whether this decreased viral replication in PBMCs is caused by, for instance, triggering of an antiviral response in these cultures remains to be determined. Indeed, macrophages and DCs present in PBMCs from *MAVS* minor donors may exert a DDX3-induced MAVS-mediated type I IFN response upon in vitro HIV-1 infection, which can interfere with virus replication in the CD4^+^ T cells. Besides functioning as cytoplasmic sensors of viral RNA, DDX3 and RIG-I have the ability to sense viral RNA in the nucleus [75,76,77]. It is unknown whether *MAVS* genetic variation affects intracellular localization of DDX3 and RIG-I and subsequent antiviral type I IFN responses. However, direct infection levels of CD4^+^ T cells did not differ between *MAVS* genotypes, suggesting that the decreased infection of CD4^+^ T cells in the PBMC culture is not due to intrinsic CD4^+^ T cell factors. These data suggest that the protective effect of the *MAVS* minor genotype is associated with the induction of a local antiviral response in DC and monocyte/macrophages, which results in decreased susceptibility of CD4^+^ T cells to HIV-1 infection and, thus, leading to the observed decrease in CD4^+^ T cell-associated proviral DNA in vivo.

In conclusion, individuals carrying the *MAVS* minor genotype show no significant decrease of CD4^+^ T cell counts during a 7-year follow up, and harbored significantly lower levels of infected CD4^+^ T cells, whereas no differences in the level of immune activation and virus-specific immune responses were observed. In vitro infection assays demonstrated that PBMCs obtained from donors carrying the *MAVS* minor genotype were less susceptible to HIV-1 replication. Although the precise underlying mechanism remains unclear, our data suggest that the protective effect of the *MAVS* minor genotype may be exerted by the initiation of local innate responses affecting viral replication and CD4^+^ T cell susceptibility.

## Figures and Tables

**Figure 1 viruses-12-00764-f001:**
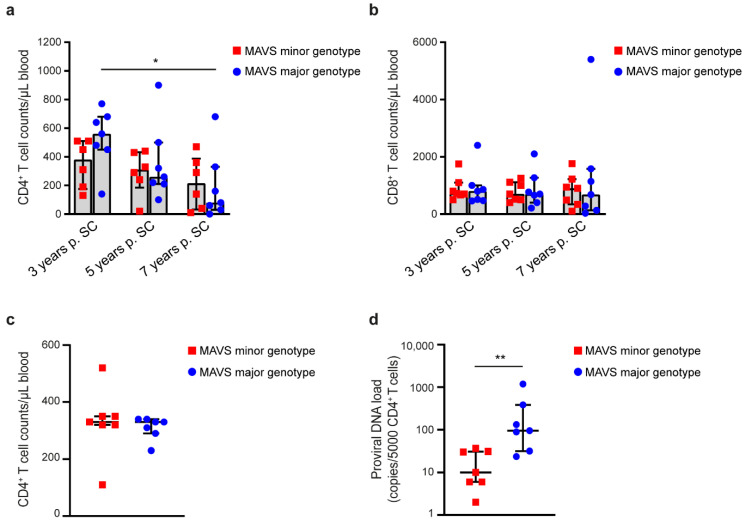
*MAVS* minor genotype is associated with sustained CD4^+^ T cell counts and a decreased level of cell-associated proviral DNA load during human immunodeficiency virus 1 (HIV-1) infection. (**a**) CD4^+^ T cell counts were assessed at 3, 5, and 7 years post seroconversion (p.SC) in untreated HIV-1-infected individuals carrying a *MAVS* minor or *MAVS* major genotype. (**b**) CD8^+^ T cell counts were assessed at 3, 5, and 7 years p.SC in individuals carrying a *MAVS* minor or *MAVS* major genotype. (**c**) CD4^+^ T cell counts and (**d**) the number of HIV-1 DNA copies per 5000 CD4^+^ T cells using qPCR were obtained from untreated HIV-1-infected individuals carrying the *MAVS* minor or *MAVS* major genotype 3.5–8.5 years p.SC. Each square or dot represents a different study participant (median ± IQR). All significant differences are indicated: * *p* < 0.05, ** *p* < 0.01, paired Wilcoxon test (**a**) and unpaired Mann–Whitney test (**d**).

**Figure 2 viruses-12-00764-f002:**
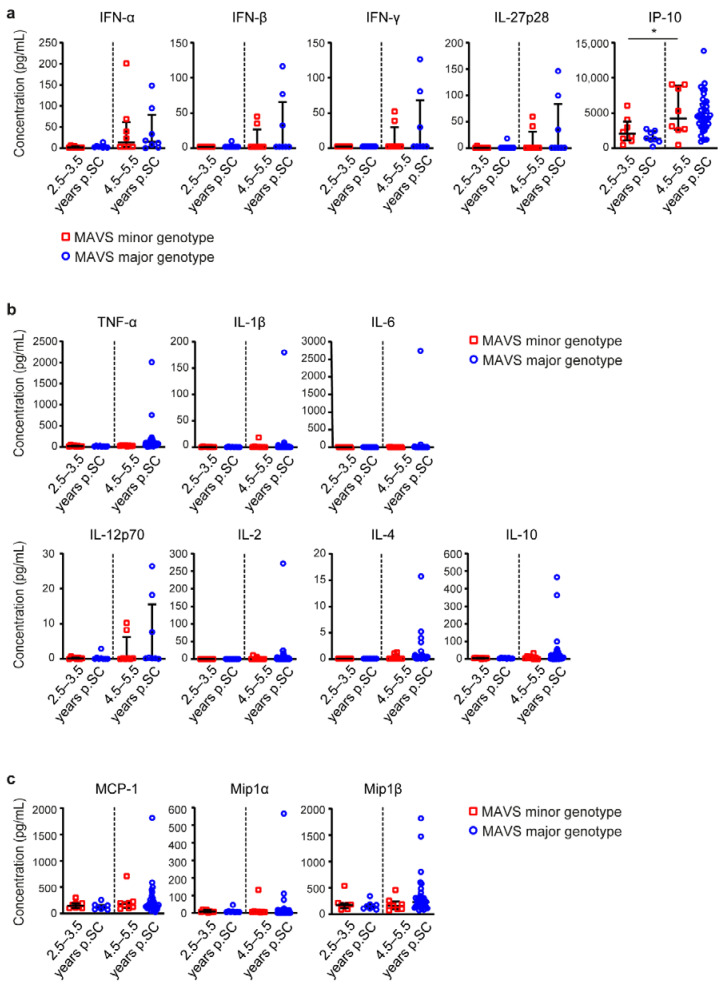
*MAVS* genetic variation does not affect immune activation levels. (**a**) Concentrations of interferon (IFN) and IFN-induced molecules in serum samples obtained 2.5–3.5 and 4.5–5.5 years p.SC of untreated HIV-1-infected individuals containing *MAVS* minor or *MAVS* major genotype were measured using a multiplex assay. (**b**) Concentrations of pro-inflammatory, regulatory, and anti-inflammatory molecules and (**c**) chemokines in serum samples obtained 2.5–3.5 and 4.5–5.5 years p.SC of untreated HIV-1-infected individuals containing *MAVS* minor or *MAVS* major genotype were measured using a multiplex assay. Each square or dot represents a different study participant (median ± IQR.). All significant differences are indicated: * *p* < 0.05, paired Wilcoxon test.

**Figure 3 viruses-12-00764-f003:**
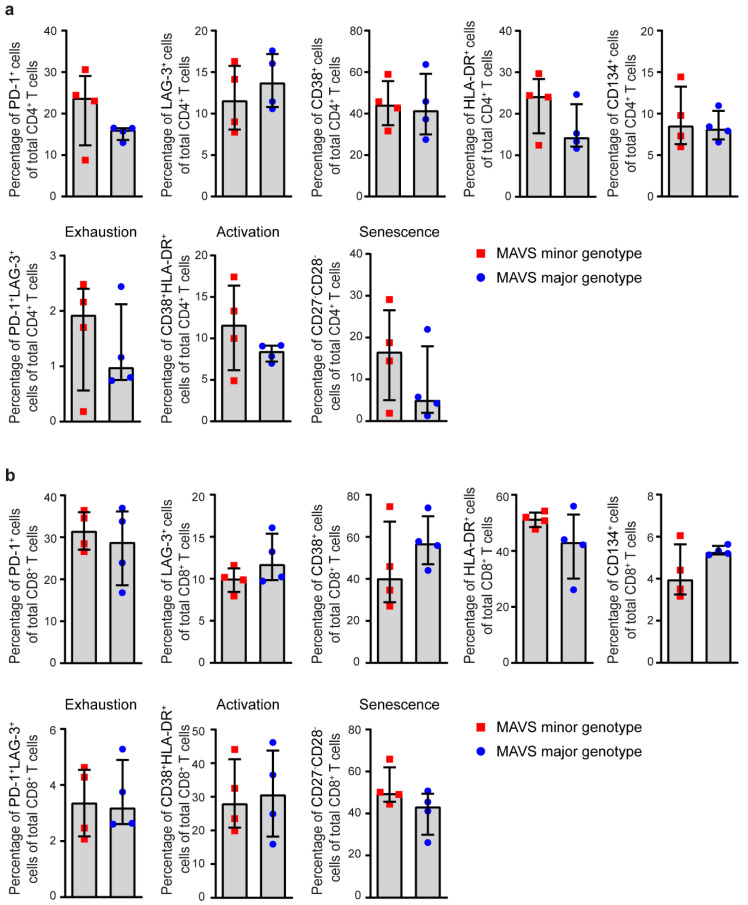
*MAVS* genetic variation does not affect T cell exhaustion, activation, and senescence. (**a**) Percentages of PD-1^+^, LAG-3^+^, CD38^+^, HLA-DR^+^, CD134^+^, exhausted (PD-1^+^LAG-3^+^), activated (CD38^+^HLA-DR^+^), and senescent (CD27^−^CD28^−^) cells within CD4^+^ and (**b**) CD8^+^ T cells of untreated HIV-1-infected individuals with a *MAVS* minor or *MAVS* major genotype 2.5–3.5 years p.SC were analyzed using flow cytometry. Each square or dot represents a different study participant (median ± IQR). No significant differences between HIV-1-infected individuals with a *MAVS* minor or *MAVS* major genotype were observed.

**Figure 4 viruses-12-00764-f004:**
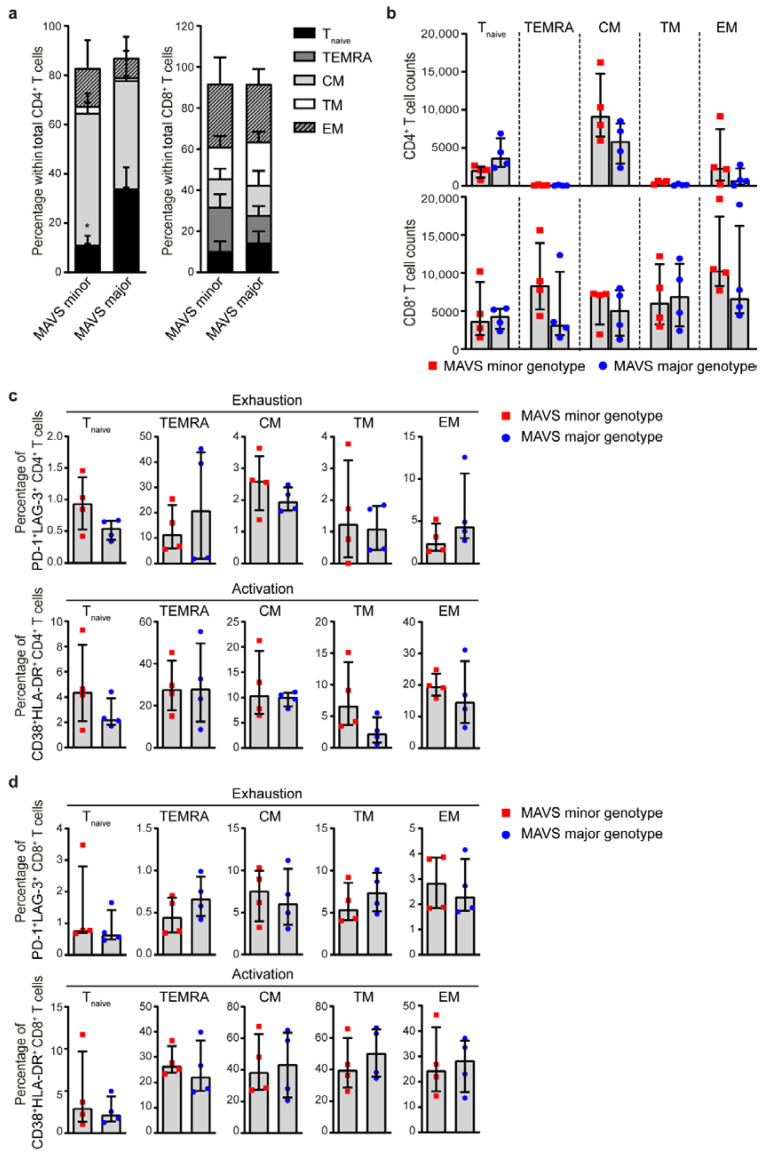
*MAVS* minor genotype is associated with a decreased percentage of naïve CD4^+^ T cells. (**a**) Percentages and (**b**) cell counts of naïve (T_naive_; CD45RA^+^CD27^+^CCR7^+^), terminally differentiated effector memory (TEMRA; CD45RA^+^CCR7^−^CD27^−^), central memory (CM; CD45RA^−^CCR7^+^CD27^+^), transitional memory (TM; CD45RA^−^CCR7^−^CD27^+^), and effector memory (EM; CD45RA^−^CCR7^−^CD27^−^) cells within CD4^+^ and CD8^+^ T cells were analyzed using flow cytometry. (**c**) Percentages of exhausted (PD-1^+^LAG-3^+^) and activated (CD38^+^HLA-DR^+^) CD4^+^ T cells and (**d**) CD8^+^ T cells within T_naive_, TEMRA, CM, TM, and EM populations were analyzed using flow cytometry. Each square or dot represents a different study participant (median ± IQR). All significant differences are indicated: * *p* < 0.05, unpaired Mann–Whitney test.

**Figure 5 viruses-12-00764-f005:**
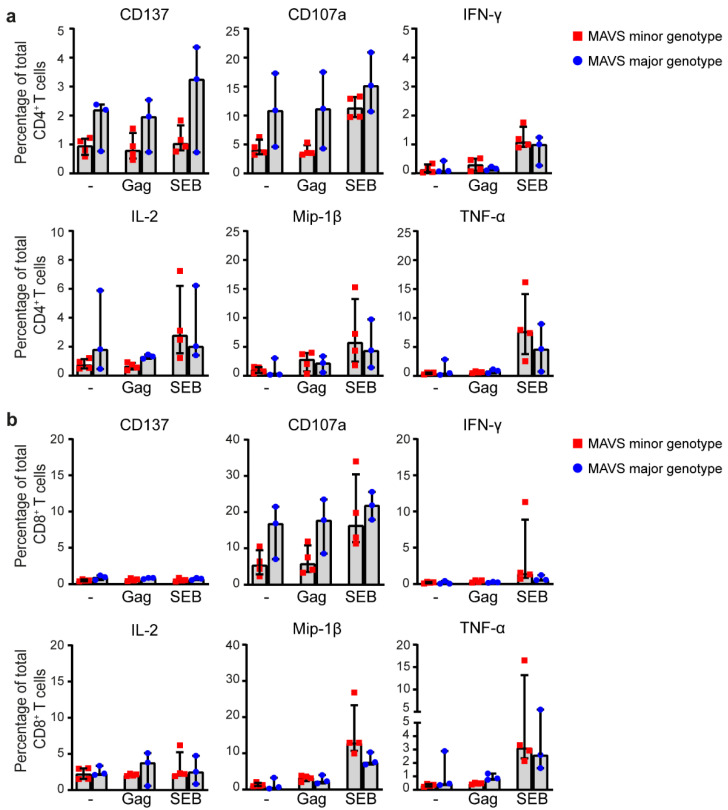
*MAVS* genetic variation does not affect intracellular cytokine levels of HIV-1-specific T cells. (**a**) PBMCs of individuals carrying a *MAVS* minor or *MAVS* major genotype obtained 2.5–3.5 years p.SC were left unstimulated (-), or were treated with HIV-1 consensus B Gag peptide pool (Gag) or staphylococcal enterotoxin B (SEB). After 6 h, the percentages of surface molecules expressing CD137^+^ and CD107a^+^ and intracellular expression of IFN-γ^+^, IL-2^+^, Mip-1β^+^, and TNF-α^+^ CD4^+^ T cells and (**b**) CD8^+^ T cells were analyzed using flow cytometry. Each square or dot represents a different study participant (median ± IQR). No significant differences between HIV-1-infected individuals with a *MAVS* minor or *MAVS* major genotype were observed.

**Figure 6 viruses-12-00764-f006:**
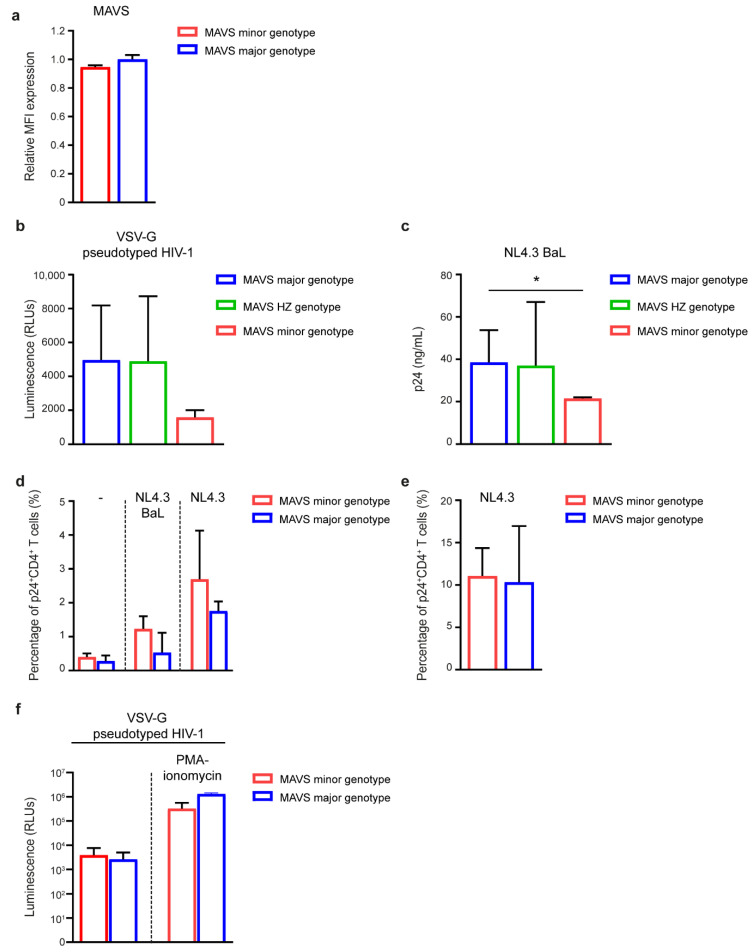
*MAVS* genotype affects viral replication in vitro. (**a**) Relative mean fluorescent intensity (MFI) of intracellular MAVS protein expression in monocyte-derived dendritic cells determined by flow cytometry. (**b**) PBMCs of healthy donors with either a *MAVS* minor, *MAVS* heterozygous (HZ), or *MAVS* major genotype were infected with single round VSV-G-pseudotyped HIV-1 containing a fire-fly luciferase gene. After 72 h, luminescence was measured by relative light units (RLUs). (**c**) PBMCs of healthy donors with the different *MAVS* genotypes were infected with R5 tropic strain NL4.3 BaL. After 8 days, virus production was determined by the detection of Gag p24 in the supernatant using ELISA. (**d**) CD4^+^ T cells of healthy donors with the different *MAVS* genotypes were not infected, infected with R5-tropic strain NL4.3 BaL or X4-tropic strain NL4.3 or (**e**) pre-treated with phorbol myristate acetate (PMA) and ionomycin followed by NL4.3 infection. After 8 days, virus replication was determined by the detection of intracellular Gag p24 using flow cytometry. (**f**) CD4^+^ T cells of healthy donors with the different *MAVS* genotypes were pre-treated with or without PMA and ionomycin followed by infection with single round VSV-G-pseudotyped HIV-1 containing a fire-fly luciferase gene. After 72 h, luminescence was measured and results are given as relative light units (RLUs). Data are representative of four (**a**), 25 (**b**), 24 (**c**) or three (**d**–**f**) *MAVS* major individuals, five (**b**,**c**) *MAVS* heterozygous individuals and four (**a**), two (**b**–**f**) *MAVS* minor individuals (median ± IQR). All significant differences are indicated: * *p* < 0.05, unpaired Mann–Whitney test.

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
