# Peer review of "MAVS Genetic Variation Is Associated with Decreased HIV-1 Replication In Vitro and Reduced CD4+ T Cell Infection in HIV-1-Infected Individuals"

_viruses, 2020, doi:10.3390/v12070764_

Round 1

Reviewer 1 Report

The manuscript of Stunnenberg and co-authors focuses on analysis of the effect of the MAVS minor genotype on innate and adaptive immune responses to HIV-1 infection.  This study is a continuation of the earlier work of the same research group published in Nature Immunology (Gringhuis et al. Nat Immunol, 2017, 18:225-235) where the authors revealed that HIV-1-induced activation of the mitotic kinase PLK1 in dendritic cells suppressed the MAVS signaling, triggered in turn by the binding of HIV-1 RNA to DDX3 helicase, and therefore led to reduced IFN-I and cytokine response to infection.   Meantime, a rare dual MAVS (Q198K,S409F) mutant displayed resistance to PLK1 inactivation and hence contributed to the reduced viral RNA load.  In present study the authors investigated the major mechanisms related to immune activation, which might be responsible for the effect of this MAVS genetic variation on control of HIV-1 during the asymptomatic phase of infection.  Analysis of CD4+ T cells counts, the levels of immune activation, T cell exhaustion, activation, senescence and intracellular cytokine responses of HIV-1-specific T cells did not reveal any significant differences between individuals with the wild-type and mutant MAVS genotypes.  However, the individuals homozygous for the MAVS minor genotype had lower levels of CD4+ T cell-associated proviral DNA.  The authors found a slight decrease in the proportion of naïve CD4+ T cells in HIV-1-infected individuals with the MAVS minor genotype.  But in Discussion they suggested that this observation may be a result of preservation of the total memory/effector CD4+ T cell population (lines 389-390).  Finally, the experiment with in vitro infection of PBMCs of healthy donors carrying the MAVS minor genotype with VSV-G pseudotyped HIV-1 or R5 tropic NL4-3Bal showed significantly decreased levels of viral replication as compared with the infected cells isolated from the healthy donors with MAVS heterozygous and MAVS major genotypes.

Thus, the proposed manuscript clearly shows that the effect of the MAVS minor genotype on HIV-1 viral RNA load and viral replication in general is not related (or less related) to IFN-I and cytokine response, and as a whole to the immune activation and virus-specific immune responses.  Indeed, in the conclusion the authors say that “Although the underlying mechanism remains unclear, their data suggest that the protective effect of the MAVS minor genotype is exerted by inhibiting HIV-1 replication in CD4+ T cells” (lines 80-81).  It should be noted here, that in Discussion the authors propose, based on the results of in vitro infection of PBMCs, that decreased viral replication could be related to an unknown CD4+ T cell intrinsic factor or caused by triggering of an antiviral response (lines 405-407).  This hypothesis, as well as the results shown in the current version of the manuscript indicate the relevance of further studies in a given direction.

The current version of the manuscript is of interest, reasonable, well and logically written, and discussing significant observations of the effect of the minor MAVS genotype of HIV-1 infection in CD4+ T cells.  In present variant the manuscript looks suitable for publication.  However, the authors do not provide the mechanism of particular action of the MAVS mutant on the viral replication, which reduces the scientific weight of this study.  The result of the last experiment suggests very interesting outcomes and carrying out of a few additional experiments clarifying potential effect of the mutant could add to understanding of this molecular mechanism.

Infection of PBMCs from healthy donors with either a MAVS minor, MAVS heterozygous (HZ) or MAVS major genotype with VSV-G pseudotyped, Luc-encoding HIV-1 revealed a dramatic difference in the luciferase expression – lower level in the cells with MAVS minor genotype.  The similar results were observed for p24 protein in PBMCs infected with R5 tropic NL4-3 (Fig. 6; lines 317-325).  Since the VSV-G pseudotyped and R5 tropic viruses exploit different mechanisms of entry and nuclear import, the results suggest that suppression of HIV replication occurred at post-entry and reverse transcription stages of infection.

Both RIG-I and DDX3 are shown to be nuclear-shuttling proteins able to sense viral RNA in the nuclear compartment (Liu et al. Nat Commun, 2018, 9:3199; Brennan et al. Eur J Cell Biol, 2018, 97(7):501-511; van Voss et al. Onco Targets Ther, 2017; 10: 3501–3513).  This suggests that MAVS mutations could be responsible for intracellular localization of these RNA sensors and binding of HIV-1 RNA in the nucleus.  This might not only induce the interferon response, but also affect nuclear export and integrity of the viral mRNA/genomic RNA.  In this context, the authors could test for (1) subcellular localization of RIG-I and DDX3 in the MAVS mutant and WT genotype cells; (2) nuclear and cytoplasmic HIV-1 RNA binding to these proteins; (3) nuclear export of spliced and unspliced HIV-1 RNA and (4) release of the viral particles (including incorporation of the viral RNA into the virions - DDX3 has been shown abundant in HIV-1 virions).  Reduced virion production can explain lower count of HIV-1 proviral DNA in CD4+ T cells with the MAVS minor genotype (as a result of reduced infection level).  Separate analysis of viral replication (R5 tropic HIV-1) in PBLs (dividing cells) and MDMs (non-dividing cells) could also be important.

Minor Essential Revisions

  1. Lines 227-228: “Furthermore, the levels of several pro-inflammatory cytokines (TNF-α, IL-1β, IL-12p70, IL-6), regulatory cytokines (IL-2, IL-4), anti-inflammatory cytokines (IL-10, IL-6) and chemokines… did not differ between the different MAVS genotypes”. IL-6 is mentioned as pro-inflammatory and anti-inflammatory cytokine in the same time.  In most cases this cytokine is considered as an inflammation and cellular senescence marker (also pro-inflammatory).

  1. Lines 319-321: “Even though virus infection of PBMCs with a MAVS major and heterozygous genotype was similar, infection levels were decreased in PBMCs of individuals with the MAVS minor genotype (Figure 6a)”. In context of my previous comments, it would be more correct to change “infection levels” to “viral replication levels”, especially for the case of infection with VSV-G pseudotyped virus.

Author Response

Please see the attachment for a point-to-point response to the reviewer's comments 

Reviewer 2 Report

The study was designed to establish a correlation between a MVAS polymorphism with HIV sensitivity. The topic of the study is of broad interest to readers and overall the experimental design is good. However, figure 6 in the ms should be improved to support the conclusion of the study. 

As far as I can tell Fig. 6 is the key figure but the way the data was represented can be and should be improved. Upon collection of cells from the donors, the authors should first check MAVS expression and see if any of the viral production difference could be explained by difference in MAVS expression levels. Secondly, the size of the error bar of MVAS major group in figure 6b makes it hard to compare the difference between MAVS major and minor. What was the N number for all groups? Was MAVS minor’s error bar much smaller because a different N was used? The N number is missing so it is difficult for me to make a conclusion with the current data set. Perhaps a different method can be used to quantify p24?  An alternative way is to titer the outcome viruses if p24 ELISA or WB is not quantitative enough.  Since this is one of the most important experiment in the study, the design should be optimized and data more cautiously presented. 

Author Response

Please see the attachment for a point-to-point response to the reviewer's comments.

Reviewer 3 Report

This study by Stunnenberg et al. seeks to find a mechanistic basis for their previous observation of decreased set point viral loads in individuals with a certain mutation in MAVS. To this end they compared cell-based characteristic of individuals of two different genotypes, mainly looking at blood/PBMCs. Most of their data points to the notion that T cell immune compartment and their functionality is similar in both genotypes except that PBMCs from minor genotype individuals may be resistant to HIV-1 to some extent. Whether this observation alone would explain the set point viral load difference is not clear.

Additional comments:

  1. Please perform stringent statistical testing to all data presented and indicate them appropriately with in each figure, wherever applicable. At several instances data look (significantly) different but it is not indicated, whether that is the case or not.
  2. Can authors please explain why they use mean instead of median for most plots? Do they assume normal distribution of data with so few data points (generally 3-5 donors per experiment)?
  3. For Fig. 4, authors should indicate the absolute cells numbers for each T cell subset, as percentages can be misleading in this case. Secondly, I do not agree with authors’ claim that only difference between these MAVS genotypes is Tn proportions. Clearly TEM and TM subsets have also changed dramatically, although it’s hard to quantitate from the plots, due to the way they are plotted. The trend appears to be statistically significant. They should increase number of donor individuals to confirm this observation.
  4. Baseline levels of CD137, CD107a and IL2 are lower by half or more between two genotypes (Fig. 5), is it biologically relevant or an experimental artifact?
  5. It was a bit disappointing to note that authors didn’t pursue the most interesting result in their manuscript, which is that PBMCs from MAVS minor genotype appear to show lower viral replication rates. They should delve deeper into this observation, what’s the mechanism behind this observation? For example, which cell types are preferentially infected or not, among these two MAVS genotypes and so on.

Author Response

(The authors gave the same response as above.)

Round 2

Reviewer 2 Report

all points have been addressed, I do not have further comments. 

Reviewer 3 Report

I am satisfied with their rebuttal.